# Cashew (*Anacardium occidentale*) Nut-Shell Liquid as Antioxidant in Bulk Soybean Oil

**DOI:** 10.3390/molecules27248733

**Published:** 2022-12-09

**Authors:** Sandra-Yaneth Gaitán-Jiménez, Luz-Patricia Restrepo-Sánchez, Fabián Parada-Alfonso, Carlos-Eduardo Narváez-Cuenca

**Affiliations:** Universidad Nacional de Colombia, Sede Bogotá, Departamento de Química, Facultad de Ciencias, Food Chemistry Research Group, Carrera 45 No 26-85, Bogotá 111321, Colombia

**Keywords:** *Anacardium occidentale*, natural antioxidants, cardanols, anacardic acids, cardols

## Abstract

Recently, natural antioxidants for the food industry have become an important focus. Cashew nut-shell liquid (CNSL) is composed of compounds that can act as natural antioxidants in food systems. The aim of this work was to evaluate the potential of CNSL and its components to act as natural antioxidants in a bulk oil system. CNSL was treated with calcium hydroxide to obtain two fractions [cardol/cardanols acid fraction (CCF) and anacardic acid fraction (AF)]. CNSL, FF and AF were analyzed by thin-layer chromatography and Fourier-transform infrared spectroscopy. The protective effects of CNSL, CCF and AF were tested in terms of the peroxide value of bulk soybean oil in accelerated assays and were compared against controls with and without synthetic antioxidants (CSA and CWA). CNLS, CCF, AF and CSA were tested at 200 mg/kg soybean oil by incubation at 30, 40, 50 and 60 °C for five days. The activation energy (E_a_) for the production of peroxides was calculated by using the linearized Arrhenius equation. Thin-layer chromatography and Fourier-transform infrared spectroscopy revealed that (i) CNSL contained cardanols, anacardic acids, and cardols; (ii) CCF contained cardanols and cardols; and (iii) AF contained anacardic acids. CSA (E_a_ 35,355 J/mol) was the most effective antioxidant, followed by CCF (E_a_ 31,498 J/mol) and by CNSL (E_a_ 26,351 J/mol). AF exhibited pro-oxidant activity (E_a_ 8339 J/mol) compared with that of CWA (E_a_ 15,684 J/mol). Therefore, cardols and cardanols from CNSL can be used as a natural antioxidant in soybean oil.

## 1. Introduction

Antioxidants from natural sources are of interest in the food industry for controlling the stability of bulk edible oils during processing, storage and use. This trend has emerged as an effort to replace synthetic antioxidants due to safety concerns, including liver damage and involvement in carcinogenesis [1]. For instance, butyl hydroxyanisole (BHA), a commonly used synthetic antioxidant, can act in a synergistic manner with other carcinogenic agents, enhancing their effect [1].

The effectiveness of a natural antioxidant can be tested, ranked and compared against a synthetic antioxidant, e.g., by means of accelerated oxidation storage assays, in which edible oils are stored at temperatures higher than room temperature to induce degradation in relatively short times [2]. During an accelerated oxidation storage assay of an edible oil, a crude extract, fractions of them or pure compounds (synthetic antioxidants or natural ones) are added, the mixture is incubated at a fixed temperature and then physicochemical parameters are evaluated according to time [2,3].

Worldwide, in 2020, a total area of 7.1 million ha was dedicated to the production of cashew nuts (Anacardium occidentale), with 4.2 million tons cashew nuts being produced [4]. The nut is commercialized by the food industry, and the cashew nut-shell liquid (CNSL) is a byproduct [5]. CNSL can be obtained, e.g., by extrusion, roasting or solvent extraction [6]. Among phenolic compounds present in solvent-extracted CNSL (Figure 1), anacardic acids are by far the most abundant (58–64% *w*/*w*), followed by cardols (20–22% *w*/*w*) and cardanols (2–10% *w*/*w*) [7,8]. The aforementioned phenolic compounds are recognized because of their remarkable high antioxidant activity as tested by different assays [9,10,11].

Experiments conducted at high temperature (180–240 °C) in edible oils have demonstrated that CNSL has a similar effect to those of BHA, BHT and TBHQ towards the control on the peroxide value in corn oil. The measurement of peroxides is interesting because it measures primary oxidation in oil systems. Moreover, CNSL was found to be more effective than tert-butyl hydroquinone (TBHQ) for inhibiting the *trans* isomerization reaction of corn, soybean and sunflower oils [9]. Likewise, cardols and cardanols are better antioxidants than butylated hydroxy toluene (BHT) of a biodiesel, as shown by both electrochemical and Rancimat assays [10]. Cardanols and cardols have higher antioxidant activity as compared with that of the synthetic antioxidant BHT, as judged by the medium inhibitory concentration (IC50) values when measured by the ABTS and the DPPH methods [11]. In contrast, those natural compounds have similar antioxidant activity to that observed for BHT when measured by the DPPH method [11]. Although both methods, ABTS and DPPH, are based on the radical scavenging capacity of antioxidants by electron transfer reactions, the observed differences could be related to intrinsic differences in the molecules used as sources of free radicals. Furthermore, as shown by the aforementioned authors, anacardic acids are the least effective antioxidant phenolics from CNSL when measured by the ABTS and DPPH methods. A recent study evaluated the effect of different extraction solvents towards the extraction of phenolic compounds, as measured by the Folin–Ciocalteau strategy [12]. In such research, extracts were characterized by mass spectrometry, and differences in chemical composition were found because of the type of extraction solvent. No further attempts, nevertheless, were made to evaluate the antioxidant activity of extracts with different phenolic compositions.

Besides a previous report [9], no further information on the use of CNSL or its fractions for the stability of edible oils is available in literature. The aim of this research was, therefore, to evaluate the effect of CNSL and some of its fractions on the stability of soybean oil in accelerated oxidation assays and to rank them against a commercial synthetic antioxidant mixture.

## 2. Results and Discussion

### 2.1. Composition of CNSL and Its Fractions

CNSL was obtained with a yield of 34.7% (*w*/*w*), a value that agrees with previous reports (34.3–35.4% *w*/*w*) [8]. TLC revealed that CNSL was composed of cardols (Rf 0.40), anacardic acids (Rf 0.58–0.65) and cardanols (Rf 0.75). Anacardic acids were the main compounds present in AF, whereas cardols and cardanols were the main compounds present in CCF, as observed by TLC. The elution order agreed with the polarity of the phenolic compounds present in CNSL: Cardols, followed by anacardic acids and by cardanols.

Common features were observed for CNSL, AF and CCF when analyzed by FTIR (Figure 2). The presence of phenolic hydroxyls was verified by peaks of around 3500–3371 cm^−1^ due to O-H stretching vibrations and by peaks at 1301–1303 and 1246 cm^−1^, corresponding to the O-H flexion and C-O tension of the phenolic group, respectively. In addition, the signals around 3010, 2926 and 2854 cm^−1^ can be assigned to the C-H stretching of aromatic groups, -CH_2_ aliphatic groups and -CH_3_ aliphatic groups, respectively. The signal that appears at 1599–1607 cm^−1^ can be assigned to the vibration of the aromatic -C=C linkages. Furthermore, the signal observed in CNSL and AF, which was absent in CCF, at 1644 cm^−1^ can be assigned to the C=O vibration of carboxylic acids. The FTIR spectra of CNSL was in agreement with that described previously [13]. The TLC and FTIR spectra suggested that the main components of CNSL extract are cardols, anacardic acids and cardanols, with anacardic acids as the main component of the AF fraction and cardols and cardanols as the representative components of the CCF fraction.

### 2.2. Accelerated Storage Experiments

Before accelerated experiments were conducted, a peroxide value of 0.2 ± 0.0 meq/kg in the soybean oil without antioxidants (CWA) was obtained. This value is lower than that reported for refined soybean oil (0.6–1.7 meq/kg) [14,15,16,17].

As a result of the accelerated storage measurements, the peroxide value was plotted as a function of storage time (up to 5 days) at 30, 40, 50 and 60 °C (Figure 3). The peroxide value in CWA increased linearly with the storage time at each of the tested temperatures. Thus, no initiation (or induction) period was observed in CWA. In contrast, an initiation period for soybean oil was reported to range from 96 days, if stored at 25 °C, to 5.3 days, if stored at 60 °C [16].

Opposite to CWA, samples containing CNSL, AF, CCF or CSA exhibited two distinguishable phases: the first one occurred at low rate, from 0 to 24 h (induction phase), and the second one occurred at higher rate, from 24 to 120 h (propagation phase). Once the induction time was finished, a linear relationship between the peroxide value and storage time was observed at any of the tested temperatures, and a pseudo zero-order reaction kinetic was adjusted [18], with the *k* value being the slope of such straight lines. As the *k* value becomes increased, the oxidation rate increased. For any of the treatments (CNSL, AF, CCF, CSA and CWA), the *k* values increased as the incubation temperature was increased (*p* ≤ 0.05). Furthermore, at 30 and 40 °C, the following *k* values order was obtained: AF > CWA > CNSL > CCF > CSA, whereas at 50 and 60 °C, the order was AF > CWA ≈ CNSL > CCF ≈ CSA.

E_a_ can be interpreted as a measure of the feasibility at which an edible oil is oxidized; as the E_a_ decreases, the oil becomes more susceptible to oxidation. When calculating the activation energies (Figure 4) for the propagation stage, the E_a_ values were found to follow the following order: AF (8339 ± 645 J/mol, relative standard deviation RSD = 7.7%) > CWA (15,684 ± 107 J/mol, RSD = 0.7%) > CNSL (26,351 ± 1536 J/mol, RSD = 5.8%) > CCF (31,498 ± 68 J/mol, RSD = 0.2%) > CSA (35,355 ± 529 J/mol, RSD = 0.1%). In all cases, differences in the E_a_ among treatments were statistically significant (*p* ≤ 0.05). From the E_a_ values, it can be stated that CCF was the most effective among the natural antioxidants tested, and AF promoted the oxidation of the oil.

The E_a_ for the same type of oil without added antioxidants (CWA; 15,684 ± 107 J/mol) was lower than that reported previously (736,384 J/mol) [16]. This result, together with the lack of an observable initiation period obtained in CWA might be related to the composition of the soybean oil. It has been shown that the presence of iron, monoolein or free stearic acid increases the rate of soybean oil oxidation [19,20]. Furthermore, fatty acid composition and structural characteristics were reported to affect the stability of edible oils to oxidation [21]. For example, both monounsaturated and polyunsaturated fatty acids are more susceptible to be oxidized as compared with saturated fatty acids [22].

The highest effectiveness of CCF as compared with CNSL for inhibiting the oxidation of soybean oil might be related to its composition. Cardanols and cardols present in CCF are better antioxidants as compared with anacardic acids (present in CNSL) [23]. Furthermore, anacardic acids were more concentrated in AF than they were in CNSL. CNSL exhibited medium antioxidant activity, whereas AF exhibited pro-oxidant activity. We suggest, therefore, that anacardic acids are the responsible compounds for the pro-oxidant behavior of the fraction. The pro-oxidant effect of anacardic acids might be related to the electron-withdrawing effect of the carboxylic acid group, which negatively affects the H-donating ability of the hydroxyl moiety attached to aromatic compounds [24]. The present research shows that, if compared at the same concentration (200 mg/L), although CNSL had a moderate protective effect towards the peroxide formation in soybean oil, the CCF fraction obtained from CNSL had a high protective effect, comparable to that exhibited by the mixture of synthetic antioxidant.

This research provides promising results for the use of natural extracts to protect edible oils. By means of accelerated assays, natural sources of antioxidants were proven to be effective on the stability of soybean oil bulk systems. A purple onion peel extract, obtained by extraction with aqueous ethanol at 25 °C, was proven to be effective to protect soybean oil from oxidation, as assessed by several parameters, including the peroxide index [25]. Similarly, extracts that were prepared from sumac (*Rhus coriara* L.) fruits with different extraction methods with aqueous ethanol as the extractant, protected soybean oil from oxidation [26].

## 3. Materials and Methods

### 3.1. Chemicals

Fresh refined soybean oil without synthetic antioxidants was obtained from Duquesa S.A. (Bogotá DC, Colombia) and was kept under darkness at 7 °C until use. Synthetic antioxidant (6% TBHQ, 14% BHT, 6% BHA, 36% monoglyceride citrate, 2% propylene glycol and 39% vegetable oil, composition expressed in *w*/*w*) was obtained from Duquesa S.A. as well. Organic solvents and chemicals were purchased either from Merck (Darmstadt, Germany) or Panreac Química SLU (Barcelona, Spain). Thin-layer chromatography (TLC) 60 G plates were bought from Merck.

### 3.2. Plant Materials

Cashew nuts from the Criollo Llanero variety were provided by AGROSAVIA from Puerto López in the municipality of Meta, Colombia (altitude 330 m above sea level, latitude 9°6′ N and longitude 73°34′ W, with an average temperature of 26 °C), and then they were transported to Bogotá D. C. AGROSAVIA is a Colombian institution which conducts research on agricultural issues. Once in the lab, the nuts were frozen at −10 °C for 12 h, and then the nut-shell was mechanically separated from the nut.

### 3.3. Cashew Nut-Shell Liquid (CNSL) Extraction

Liquid extraction from the cashew nut-shell was based on that reported elsewhere [8]. Nut-shells (300 g) were extracted by maceration with light petroleum ether 40–60 (1.0 L) for 30 h. After filtration, the nut shells were blended with light petroleum ether (100 mL) and were filtered again. Blending with fresh light petroleum ether was performed five times, each time with 100 mL. All extracts were mixed together and rota-evaporated at 40 °C (approximately 3 h). The free-solvent extract (104 g, 34.7% *w*/*w* yield) was the solvent-extracted cashew nut-shell liquid (CNSL). CNSL was stored at 4 °C in darkness until its fractionation.

### 3.4. CNSL Fractionation

The fractionation of CNSL was based on previous reports [7,27]. CNSL (100 g) was suspended in a light petroleum ether:ethanol mixture (100 mL, 2:1, *v*/*v*), shaken (200 rpm, 20 °C, 5 min) and left to decant for 30 min. The suspended CNSL was filtered, the solid was discarded and then the filtrate was rota-evaporated at 40 °C (approximately 5 h). A sample of the rota-evaporated material (50 g) was dissolved into 5% (*v*/*v*) aqueous methanol (100 mL), and then calcium hydroxide (25 g) was added. The suspension was stirred for 3 h at 50 °C, cooled down to room temperature and filtrated. Both retentate (A) and filtrate (B) were further purified.

The retentate (A) was washed five times with 20 mL ethyl acetate and was dried at 40 °C overnight. The dried material was suspended in 5% (*v*/*v*) aqueous methanol (200 mL), mixed with 37% (*w*/*w*) hydrochloric acid (80 mL) and stirred for 1 h at room temperature. This solution was partitioned twice with ethyl acetate (80 mL each) in a separation funnel and the organic layer was washed three times with distilled water (80 mL each). Powdered anhydrous sodium sulfate was then added to the organic fraction. After filtration, the organic solvent was removed by rota-evaporation at 30 °C (approximately 3 h) to obtain an oily fraction (AF, 34.6 g).

The filtrate (B) was treated with 10% (*w*/*v*) aqueous sodium hydroxide (200 mL) and was stirred for 15 min. This solution was partitioned three times with light petroleum ether-ethyl acetate (9:1, *v*/*v*, 60 mL each) in a separation funnel. The light petroleum ether-ethyl acetate fraction was washed twice with 5% (*v*/*v*) aqueous hydrochloric acid (60 mL each). Then, it was washed twice with distilled water (60 mL each). The light petroleum ether-ethyl acetate, free of polar compounds, was dried with powdered anhydrous sodium sulfate and was rota-evaporated at 30 °C (approximately 30 min) to obtain an oily fraction (CCF, 13.5 g).

### 3.5. TLC and Fourier-Transform Infrared Spectroscopy (FTIR)

The composition of CNSL, AF and CCF was monitored by TLC and FTIR. TLC was performed on silica gel F254 plates and was eluted with a petroleum ether:ethyl acetate mixture (7:3, *v*/*v*). FTIR spectra were recorded on a FTIR Perkin Elmer Paragon 500 spectrometer (Perkin Elmer, Waltham, MA, USA) using a sample film on a KBr pellet. The FTIR spectrum was recorded from 400 to 4000 cm^−1^.

### 3.6. Accelerated Oxidation Assays

The effect of CNLS, AF and CCF towards the peroxide value of soybean oil was evaluated at 200 mg/kg oil by incubation at 30, 40, 50 and 60 °C for five days. A positive control containing 200 mg/kg of synthetic antioxidant (CSA) was tested. CNLS, AF, CCF and CSA were dissolved in 1 mL of absolute ethanol to reach a concentration of 20,000 mg/L, and 200 µL was added to 20 mL of fresh refined soybean oil without synthetic antioxidants. Under those conditions, the final concentration of CNLS, AF, CCF and CSA was, in each case, 200 mg/L. After that, formulations were mixed for 5 min in an ultrasonic water bath [28]. A negative control without any antioxidant (CWA) was prepared by adding 200 µL of absolute ethanol to 20 mL of fresh refined soybean oil, following the procedure described above. Samples were stored in an oven at the programmed temperature (30, 40, 50 and 60 °C) under darkness in 50 mL amber glass bottles (surface area exposed to the air interface 3 cm^2^). Every 2 h, samples were vortexed for 10 s. Samples were analyzed for their peroxide values at 0, 6, 12, 24, 36, 48, 60, 72, 84, 96, 108 and 120 h. Experiments were conducted in triplicate. Peroxide values were plotted as a function of storage time. The slope of the straight lines represents the oxidation rate constant (*k*). Because the *k* values were measured at several temperatures, it was possible to calculate activation energy by using the linearized Arrhenius equation (Equation (1)) [16,29].
(1)$$\mathrm{Ln}\;k=-\frac{E_{a}}{\mathrm{RT}}+\mathrm{Ln}\;A$$
where *k*, E_a_, R, T and A are the oxidation rate constant, activation energy, gas constant, absolute temperature and the Arrhenius pre-exponential factor, respectively.

### 3.7. Peroxide Value

The peroxide value was measured according to the volumetric method, as described in the AOCS official method Cd 8–53 [30]. Briefly, a 5 g oil sample was mixed with a 30 mL acetic acid:chloroform mixture (3:2, *v*/*v*) and 0.5 mL of aqueous saturated potassium iodine. The produced triodide (I_3_^−^) was titrated with 0.01 M of sodium thiosulphate in the presence of 1% (*w*/*w*) aqueous starch solution as an indicator. Results were expressed as meq peroxide/kg oil.

### 3.8. Statistical Analysis

The statistical analyses were performed using Statgraphics Plus^®^ 5.1 for Windows (Statpoint Technologies Inc., Warrenton, VA, USA). The mean values and their standard deviations were reported. A one-way analysis of variance was performed. Comparisons among the means were performed by Tukey’s test (*p* ≤ 0.05).

## 4. Conclusions

From the calculated activation energies for peroxide formation, it was found that cashew nut-shell liquid (CNSL; 26,351 ± 1536 J/mol), containing, cardols, cardanols and anacardic acids, was less effective than the control with synthetic antioxidant (CSA; 35,355 ± 529 J/mol). Furthermore, the CCF fraction, rich in cardols and cardanols, obtained from the CNSL, can be used as a natural antioxidant in soybean oil as an alternative to a synthetic antioxidant, since its activation energy (31,498 ± 68 J/mol) was comparable to that of CSA. Further experiments testing higher concentrations of CCF might be useful to achieve greater protection of edible oils against oxidation and to test them not only for their peroxide values, but also for other quality parameters with health impacts, such controlling the formation of *trans* isomers. When considering the use of CNSL, its fractions or pure compounds, safety assays should be considered.

## Figures and Tables

**Figure 1 molecules-27-08733-f001:**
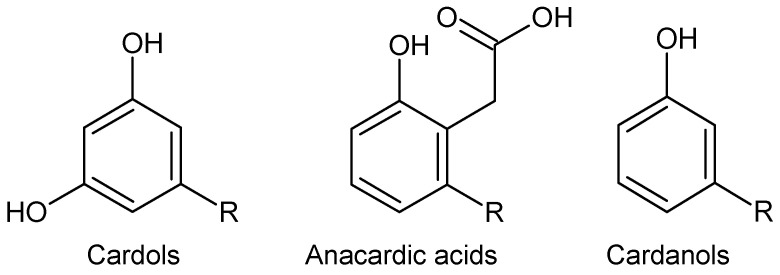
Phenolic compounds present in cashew nut-shell liquid. R = 8Z, 11Z, 14Z-Pentadecatrienyl; 8Z, 11Z-pentadecadienyl; 8Z-pentadecenyl; or pentadecyl Adapted from [7].

**Figure 2 molecules-27-08733-f002:**
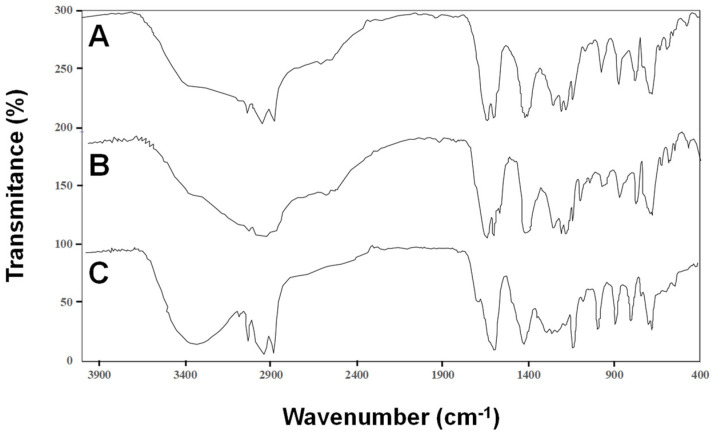
Fourier-transform infrared spectroscopy of (**A**) cashew nut-shell liquid (CNSL), (**B**) AF and (**C**) CCF. The scale was shifted upwards by 100 and 200% for AF and CNSL, respectively.

**Figure 3 molecules-27-08733-f003:**
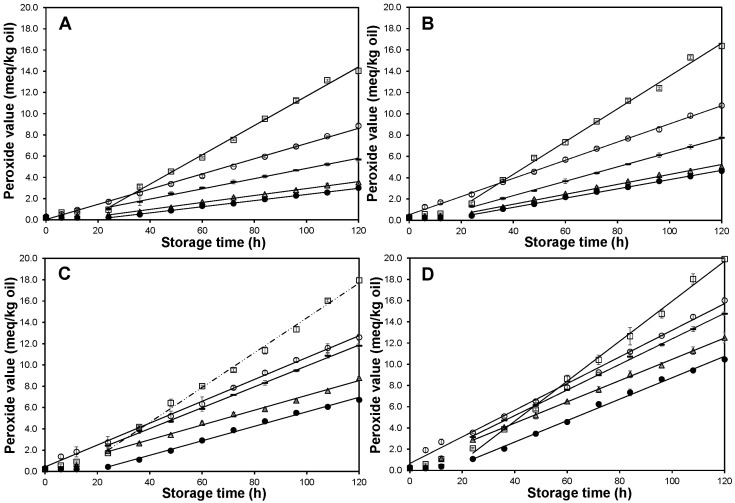
Peroxide value of soybean oil stored at (**A**) 30, (**B**) 40, (**C**) 50 and (**D**) 60 °C. Control with synthetic antioxidant, 
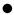
; CCF fraction, 
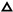
; Cashew nut-shell liquid, 

; Control without antioxidant, 
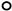
; FA Fraction, 
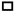
. Data are presented as mean (*n* = 3) ± SD. Only the linear part, after the induction phase was finished, was linearized. r^2^ values range from 0.996 to 0.999.

**Figure 4 molecules-27-08733-f004:**
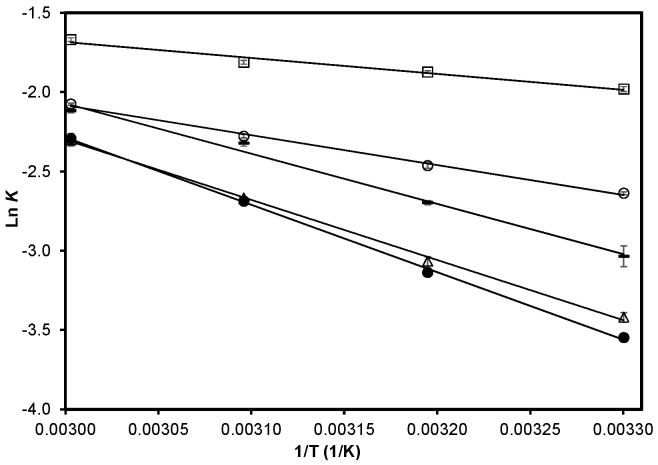
Arrhenius plot of the oxidation of soybean oil. Control with synthetic antioxidant, 
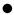
; CCF fraction, 
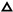
; Cashew nut-shell liquid, 

; Control without antioxidant, 
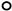
; FA Fraction, 
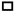
. Data are presented as mean (*n* = 3) ± SD. r^2^ values range from 0.973 to 0.999. Each slope represents E_a_/R.

## Data Availability

The data that support the findings of this study are available from the corresponding author upon reasonable request.

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
