# Peer review of "Cashew (Anacardium occidentale) Nut-Shell Liquid as Antioxidant in Bulk Soybean Oil"

_molecules, 2022, doi:10.3390/molecules27248733_

Round 1

Reviewer 1 Report

I am very grateful you for the invitation to review the manuscript molecules-2043540 by Gaitán-Jiménez and coauthors "Cashew (Anacardium occidentale) nut-shell liquid as antioxidant in bulk soybean oil”. The aim of this study was evaluated the effect of the CNSL and some of its fractions towards the stability of soybean oil in accelerated oxidation assays and to rank them against a commercial synthetic antioxidant mixture. The work is very interesting but needs adjustments to increase the quality of the material.

Comments:

- Author affiliation: Please remove the repeated address and present the corresponding email address for each author in the proper sequence.

- Abstract, Line 15: Insert reason of interest.

- Line 17: Specify what the abbreviations “FF” and “AF” refer to.

- Line 27: The claim must be better supported. Please make it clear if the results can indicate the use of the compound throughout the oil storage.

- Abstract: Please clearly specify the objective in the abstract.

- Lines 31-33: Insert information on studies of toxicity or carcinogenic potential of synthetic antioxidants, indicating potentially dangerous doses to allow comparisons.

- Lines 41-44: Indicate the proportion of waste generation in processing, presenting global data.

- Line 55-57: Please indicate the difference in antioxidant activity in relation to the tests. It needs to be clear why.

- Line 192: Include drying time.

- Line 210; 222: Check the superscript in the values.

- Materials and methods: They are adequately described and methodologically designed.

Author Response

Dear Editor,

We thank the reviewers for the time devoted to pur manuscript entitled “Cashew (Anacardium occidentale) nut-shell liquid as antioxidant in bulk soybean oil”. Here we present a response in red to each of the reviewer´s comments to the manuscript.

Reviewer 1

Comments and Suggestions for Authors

I am very grateful you for the invitation to review the manuscript molecules-2043540 by Gaitán-Jiménez and coauthors "Cashew (Anacardium occidentale) nut-shell liquid as antioxidant in bulk soybean oil”. The aim of this study was evaluated the effect of the CNSL and some of its fractions towards the stability of soybean oil in accelerated oxidation assays and to rank them against a commercial synthetic antioxidant mixture. The work is very interesting but needs adjustments to increase the quality of the material.

We would like to thank Reviewer #1 for his/her good opinion on the manuscript and the time devoted to review our work.

Comments:

- Author affiliation: Please remove the repeated address and present the corresponding email address for each author in the proper sequence.

This was corrected as requested. Lines 6-11.

- Abstract, Line 15: Insert reason of interest.

The reason of interest was included as recommended. Lines 13-14.

- Line 17: Specify what the abbreviations “FF” and “AF” refer to.

Abbreviations were defined. As a result, AF was better replaced by CCF. Then, all over the document, FF was replaced by CCF. Line 16 and all over the document.

- Line 27: The claim must be better supported. Please make it clear if the results can indicate the use of the compound throughout the oil storage.

Indeed the sentence was not clear. We included information on the type of compounds with such antioxidant activity during oil storage. Line 27.

- Abstract: Please clearly specify the objective in the abstract.

The aim of the research was included. Lines 14-15.

- Lines 31-33: Insert information on studies of toxicity or carcinogenic potential of synthetic antioxidants, indicating potentially dangerous doses to allow comparisons.

Information was introduced. No dosis for toxic effect was provided because of the synergistic effects and controversial information. Lines 33-36.

- Lines 41-44: Indicate the proportion of waste generation in processing, presenting global data.

We checked in the scientific literature and could not find specific information for the amount of waste related to the crop/production of Annacardium occidentale. If the reviewer can recommend us some references, we would be happy to include them.

- Line 55-57: Please indicate the difference in antioxidant activity in relation to the tests. It needs to be clear why.

This was solved as suggested. Lines 65-72.

- Line 192: Include drying time.

This was included as recommended in each part that rotaevaporation was done. Lines 212, 220, 231, and 239.

- Line 210; 222: Check the superscript in the values.

The superscript in cm-1 was adjusted. Line 247. Underscripts were also corrected in lines 101,

- Materials and methods: They are adequately described and methodologically designed.

No action to be taken.

Reviewer 2 Report

I read the manuscript carefully, and I can be recommended some points that need to support:

1- Please support the introduction about using natural preservative components to extend the shelf life, particularly the ones that possess the mechanism.

2- In some sections, methods need to be more clear.

3- please support the discussion with recent references supporting the aim of the study.

5 - please make a recommendation.

Author Response

Dear Editor,

We thank the reviewers for the time devoted to pur manuscript entitled “Cashew (Anacardium occidentale) nut-shell liquid as antioxidant in bulk soybean oil”. Here we present a response in red to each of the reviewer´s comments to the manuscript.

Reviewer 2

We would like to thank Reviewer #2 for his/her time invested in reviewing our manuscript. We appreciate it.

Comments and Suggestions for Authors

I read the manuscript carefully, and I can be recommended some points that need to support:

1- Please support the introduction about using natural preservative components to extend the shelf life, particularly the ones that possess the mechanism.

We had already included information on the use of cahew nut shle liquid to control the peroxide value. To emphasize more that information we reorganized the whole paragraph. Lines 58-72.

2- In some sections, methods need to be more clear.

The section Materials and Methods was revised and adjusted where needed. Each change was marked in red.

3- please support the discussion with recent references supporting the aim of the study.

We included two more recent references in the Results and discussion section. These references are related with the use of natural extracts to protect eddible oils from lipid oxidation as measured by the peroxide value and by other parameters indicative of lipid oxidation. Numbering of the references was then, modified.

Umeda, W.M. ; Jorge, N. Oxidative stability of soybean oil added of purple onion (Allium cepa L.) peel extract during accelerated storage conditions. Food Control. 2021127, 108130.

Rahmati, S.; Bazargani‐Gilani, B. ; Aghajani, N. Effect of extraction methods on the efficiency of sumac (Rhus coriaria L.) fruit extract in soybean oil quality during accelerated conditions. Food Sci. Nutr. 2022, 10, 3302-3313.

5 - please make a recommendation.

This was included at the end of the Conclusions section. Lines 294-297.

Reviewer 3 Report

The manuscript is written on a good scientific level, a mistake was probably made on the deviation of the values ​​in lines 123 and 124

 AF (8,339 ± 645 J/mol) > CWA (15,684 ± 107 J/mol) > CNSL (26,351 ± 1,536 J/mol) > 123 FF (31,498 ± 68 J/mol) > CSA (35,355 ± 529 J/mol)

Author Response

Dear Editor,

We thank the reviewers for the time devoted to pur manuscript entitled “Cashew (Anacardium occidentale) nut-shell liquid as antioxidant in bulk soybean oil”. Here we present a response in red to each of the reviewer´s comments to the manuscript.

Reviewer 3

Comments and Suggestions for Authors

The manuscript is written on a good scientific level, a mistake was probably made on the deviation of the values ​​in lines 123 and 124

 AF (8,339 ± 645 J/mol) > CWA (15,684 ± 107 J/mol) > CNSL (26,351 ± 1,536 J/mol) > 123 FF (31,498 ± 68 J/mol) > CSA (35,355 ± 529 J/mol)

We would like to thank Reviewer #3 for his/her good opinion on the manuscript. We checked the standard deviations and they are well calculated. To show in persepective such values, we calculated the relative standard deviation (RSD) and inserted those values in the manuscrip. The new versión says: AF (8,339 ± 645 J/mol, relative standard deviation RSD = 7.7%) > CWA (15,684 ± 107 J/mol, RSD = 0.7%) > CNSL (26,351 ± 1,536 J/mol, RSD = 5.8%) > CCF (31,498 ± 68 J/mol, RSD = 0.2%) > CSA (35,355 ± 529 J/mol, RSD = 0.1%). Lines 136-138.